Freeze-drying can replace cold-chains for transport and storage of fecal microbiome samples

Bensch Hanna M. hanna.bensch@lnu.se 1 2
Tolf Conny 1
Waldenström Jonas 1
Lundin Daniel 1
Zöttl Markus 1 2
1 Centre for Ecology and Evolution in Microbial Model Systems (EEMIS), Department of Biology and Environmental Science, Linnaeus University , Sweden
2 Kalahari Research Centre, Kuruman River Reserve, Van Zylsrus , South Africa
Kormas Konstantinos
Electronic publication date: 2022 Mar 15
Publication date: 2022
Volume: 10
Electronic Location ID: e13095
Received 2021 Nov 30; Accepted 2022 Feb 20
Copyright: ©2022 Bensch et al.
Copyright year: 2022
Copyright holder: Bensch et al.
License: This is an open access article distributed under the terms of the Creative Commons Attribution License, which permits unrestricted use, distribution, reproduction and adaptation in any medium and for any purpose provided that it is properly attributed. For attribution, the original author(s), title, publication source (PeerJ) and either DOI or URL of the article must be cited.
License URL: https://creativecommons.org/licenses/by/4.0/

Keywords: Microbiome, 16S, Fecal samples, DNA metabarcoding, Fukomys damarensis, Damaraland mole-rat, Amplicon, Freeze-drying

Funding: Vetenskapsrådet (2017-05296) Crafoordska Stiftelsen (2018-2259 & 2020-0976) This study was supported by grants from Vetenskapsrådet (2017-05296) and Crafoordska Stiftelsen (2018-2259 & 2020-0976). The facilities of the of the mole-rat project and the long-term study are supported by European Research Council grants 294494 and 742808 to T. H. Clutton-Brock. The funders had no role in study design, data collection and analysis, decision to publish, or preparation of the manuscript.

==============================
Background

The transport and storage of samples in temperatures of minus 80 °C is commonly considered as the gold standard for microbiome studies. However, studies conducting sample collection at remote sites without a reliable cold-chain would benefit from a sample preservation method that allows transport and storage at ambient temperature.

Methods

In this study we compare alpha diversity and 16S microbiome composition of 20 fecal sample replicates from Damaraland mole-rats (Fukomys damarensis) preserved in a minus 80 °C freezer and transported on dry ice to freeze-dried samples that were stored and transported in ambient temperature until DNA extraction.

Results

We found strong correlations between relative abundances of Amplicon Sequence Variants (ASVs) between preservation treatments of the sample, no differences in alpha diversity measures between the two preservation treatments and minor effects of the preservation treatment on beta diversity measures. Our results show that freeze-drying samples can be a useful method for cost-effective transportation and storage of microbiome samples that yields quantitatively almost indistinguishable results in 16S microbiome analyses as those stored in minus 80 °C.

Introduction

The transport and storage of microbiome samples on dry ice and in minus 80 °C freezers (i.e., super freezers) is commonly considered the gold standard (Vandeputte et al., 2017). However, for research carried out in remote locations, at field stations or when the analytical laboratory work is carried out at laboratories far away from the collection site, it can be challenging to provide super freezers and maintain uninterrupted cold chains (Kim et al., 2017; Vandeputte et al., 2017). Super freezers require a reliable source of electricity and are expensive to purchase and maintain. Ordinary freezers may be a cost efficient alternative and increasing evidence suggests that DNA storage of microbiome samples in those freezers yield similar 16S rRNA gene sequencing data profiles as samples stored in super freezers (Song et al., 2016; Gavriliuc et al., 2021). However, because the microbiota composition in frozen samples is sensitive to thawing (Cardona et al., 2012; Song et al., 2016), freezing requires an uninterrupted cold chain from sampling to library preparation and samples are therefore often transported on dry ice or in liquid nitrogen from the field sites to the laboratory. Transportation of frozen samples hence introduces the risk of thawing and incurs additional costs of cold chain logistics. This is especially problematic when the sampling location is in a remote area or in another country than the laboratory facility and commercial couriers are needed.

When freezing is not an available option, the most commonly used and tested alternative storage of samples include ethanol, RNAlater or FTA-cards (Hale et al., 2015; Song et al., 2016; Vogtmann et al., 2017; Wang et al., 2018). Although not requiring immediate freezing, these sample treatments are commonly frozen after a couple of hours or days at ambient temperatures for long-term storage prior to library preparation. Out of these alternative methods, ethanol is the cheapest and has been shown to be the most effective for preserving the microbiome community structure when freezers are not available (Hale et al., 2015). However, the use of ethanol comes with drawbacks, for example the additional step of ethanol removal prior to DNA extraction, as well as risk of evaporation. Furthermore, ethanol is both volatile and flammable, which increases shipping costs and restricts export transport options (Song et al., 2016; Vandeputte et al., 2017). Consequently, methods where samples can be transported at room temperature and without hazardous liquids could be of great value to research projects working at remote locations.

In this study, we evaluate freeze-drying as an alternative method for the preservation of fecal samples prior to transport, storage and 16S sequencing. Although freeze-drying has rarely been used in microbiome studies (Gavriliuc et al., 2021), it offers considerable advantages. No chemicals are added and the samples can be stored and transported at ambient temperature, eliminating the need for cold chain logistics and hazardous material handling. Previous studies have offered some support for this preservation technique by showing an overlap in 16S data from freeze-dried human stool samples with data from frozen samples (Kia et al., 2016) and little influence of freeze-drying on homogenized neonatal fecal samples (Shen et al., 2021). However experimental work evaluating the effects of freeze-drying on field samples and samples from non-human animals is still lacking. To evaluate whether freeze-drying, subsequent long-distance transport, and storage at ambient conditions affects the results of 16S sequencing, we conducted an experiment where we investigated the gut microbiome diversity and composition of 20 replicate fecal samples from Damaraland mole-rats (Fukomys damarensis) by comparing freeze-drying with super freezers.

Methods

Sample collection and preservation treatment

Fecal samples were collected from 20 wild-caught Damaraland mole-rats shortly after capture at the Kuruman River Reserve, South Africa, between 1st of April and 15th of October 2019 as part of a long-term study on social behaviours and ecology of this subterranean rodent (e.g., Zöttl et al., 2016; Thorley et al., 2021). The animals were from five randomly selected family groups (four individuals per group). Each sample was split into two replicate tubes with 1–3 fecal pellets per tube and stored in a minus 80 °C freezer at the field site. After all samples had been collected, one replicate per sample was thawed, and subsequently freeze-dried for 48 h at <−40 °C using an ALPHA 1-2 LDplus Freeze Dryer following the manufacturer’s protocol. Freeze-dried samples were then stored and transported at ambient temperature while frozen replicates were transported on dry ice with a commercial courier service to Sweden.

The animal captures and sample collection protocol used in this study was approved by the animal ethics committee of the University of Pretoria (EC050-16) and by Northern Cape Nature Conservation (ODB 1859-2016).

Library preparation and sequencing

The 20 samples were randomly placed on three 96-well plates together with other fecal samples from Damaraland mole-rats (as part of a larger microbiome study). Freeze-dried samples had been stored at room temperature (4 to 10 months) while frozen samples were thawed shortly before extraction. Each plate included four negative controls without a fecal sample and one mock community standard (25 µl ZymoBIOMICS Microbial Community DNA Standard). DNA from samples were extracted inside a UV-hood using the DNeasy PowerSoil Pro Kit (Qiagen) following the manufacturer’s protocol. DNA concentration was quantified using a NanoDrop™1000 spectrophotometer (Thermo Fisher Scientific), and 2.5 ng extracted DNA from each sample was amplified in an initial PCR using the primer 341F (5′-CCTACGGGNGGCWGCAG-3′) and 805R (5′-GACTACHVGGGTATCTAATCC-3′) targeting the V3-V4 region of the 16S rRNA gene (Herlemann et al., 2011; Hugerth et al. 2014). The primers included adapter sequences for Illumina n5/n7 index primers used in a second PCR. DNA samples were amplified in 25 µl reactions containing 0.5 µM of each primer and 12.5 µl Phusion High-Fidelity PCR Master Mix (Thermo Fisher Scientific). Thermal PCR cycling conditions used were as follows: 30 s at 98 °C, followed by 20 cycles consisting of 10 s at 98 °C, 30 s at 58 °C and 15 s at 72 °C, and a final 2 min elongation step at 72 °C. The PCR products were purified using AMPure XP magnetic beads (Becker Coulter, USA) and used as template in the second PCR where each sample within a sequencing plate was amplified with a unique combination of Illumina n5/n7 index primers. The 50 µl reaction mix contained 23 µl purified DNA from the first PCR, 0.2 µM index primers and 25 µl Phusion High-Fidelity PCR Master Mix. For the PCR, the following thermal cycling steps were used: 30 s at 98 °C, followed by 12 cycles consisting of 10 s at 98 °C, 30 s at 62 °C and 5 s at 72 °C, and a final 2 min elongation at 72 °C. After the second PCR and another round of purification, DNA concentrations were measured using a Qubit fluorometer (Thermo Fisher Scientific), and equimolar amounts of each sample library from individual sample plates combined into pools with a final concentration of 4 ng/µl. The pools were 300-bp paired end sequenced following standard Illumina sequencing protocols on an Illumina MiSeq platform at the Swedish National Genomics Infrastructure (NGI) at SciLifeLab in Uppsala, Sweden.

Bioinformatics and sequencing filtering

The quality of the reads was checked with FastQC v0.11.8 and MultiQC v1.9 (Andrews, 2010; Ewels et al., 2016). Raw reads from FastQ inputs were processed using the Ampliseq workflow v1.2.0dev (https://nf-co.re/ampliseq/1.2.0, Straub et al., 2020) which uses Cutadapt v.2.8 (Marin, 2011) to identify sequences with primers, remove sequences without primers and delete primers. Sequences passing the primer identification are denoised with the QIIME2 v2019.10.0 (Bolyen et al., 2019) implementation of DADA2 v.1.10.0 (Callahan et al., 2016) and Amplicon sequence variants (ASVs) are created with taxonomy assigned using the SILVA database v.132 (Quast et al., 2013) and QIIME2′s Bayesian classifier. We used the default parameters, besides specifying our own primer sequences and trimming lengths (259 for forward, and 199 for reverse reads) so that sequences were trimmed to equal lengths before the actual denoising, as suggested by the DADA2 protocol. A phylogenetic tree of the ASV sequences was estimated using SEPP on the Greengenes 16S rRNA gene reference tree (McDonald et al., 2012; Mirarab, Nguyen & Warnow, 2012; Janssen et al., 2018).

Quality check and filtering of NGS data

Analyses post Ampliseq were conducted in R version 4.0.1, using mainly functions within the tidyverse, phyloseq and vegan packages (McMurdie & Holmes, 2013; Wickham et al., 2019; Oksanen et al., 2020), and figures were created with ggplot2 (Wickham & Chang, 2016). We combined reads of samples on plates that had been sequenced twice (plate 2 and 3) and removed all contaminant ASVs identified as more prevalent in the negative control samples than true fecal samples by the prevalence method in the decontam package v1.8.0, with a threshold of 0.5 and plate number as batch argument (Davis et al., 2018).

Statistical analysis

ASV richness, Shannon index and Faith’s phylogenetic diversity (PD) were calculated on subsampled ASV counts rarefied to the minimum library size (39323 reads) with phyloseq (McMurdie & Holmes, 2013) which removed 144 ASVs. The effect of sample preservation treatment on library sizes was analysed using a linear mixed model, fitting library size as response variable, including the sample preservation treatment as fixed factor and sample identity and plate number as random factors (Bates et al., 2015). Subsequently, we tested if any of the alpha diversities were associated with sample library size using Pearson correlations. For those alpha diversity measures with a significant association between library size and alpha diversity (ASV richness and PD), we controlled for the effects of library size by calculating the residual alpha diversity from a linear regression between alpha diversity and library size. We assessed the effect of sample preservation treatment on (residual) alpha diversity measures fitting a linear mixed model with (residual) alpha diversity as response variable, the sample preservation treatment as fixed factor and sample identity and plate number as random factors. All mixed models were fitted using the R package lme4 (Bates et al., 2015). Finally, we tested Pearson correlations between alpha diversity measures from sample replicates.

To investigate beta diversity of samples, we performed two principal component analyses (PCA) with the prcomp function on two phylogeny-independent Euclidean distance matrices based on two different transformation methods of raw counts of ASVs: Hellinger transformation (Rao, 1995) using the decostand function in the vegan package (Oksanen et al., 2020) and centered log-ratio (CLR) transformation (Aitchison, 1982) using the clr function in the compositions package. These methods deal with skewed abundance distribution within amplicon microbiome data in different ways. The Hellinger transformation reduces the influence of uncommon ASVs and weighs heavier on the more common ASVs (McMurdie & Holmes, 2013), while the CLR weighs heavier on the rare ASVs (Aitchison, 1982). Additionally, we calculated weighted and unweighted UniFrac distances (Lozupone & Knight, 2005), based on a phylogeny built with SEPP (Janssen et al., 2018). Non-metric multidimensional scaling (NMDS) using the ordinate-function in the phyloseq-package was applied on UniFrac distances (McMurdie & Holmes, 2013). One sample pair which was dominated by the family Enterobacteriaceae (sample 4, see Fig. S1) strongly reduced distances between the other samples in the NMDS (see Fig. S2) and was excluded for further analysis and visualizations.

The amount of variation explained by preservation treatment and sample identity for each beta diversity measure was tested with a Permutational Multivariate Analyses of Variance (PERMANOVA) with the adonis2-function from the vegan package with sample preservation treatment, sample library size and sample identity as fixed factor, by-argument set as “margin”, and testing with and without plate number as the strata argument to control for variation between sequencing plates. Likewise for the NMDS on phylogenetic beta diversity measures, we excluded the outlier sample pair (sample 4, see Fig. S1) in the PERMANOVAs to resemble that of the visualizations of the NMDS. To evaluate multivariate homogeneity of group dispersion of sample preservation treatment and sequencing plates we ran beta disperser tests on each of the beta diversity measures.

Correlations of relative abundances of ASVs within sample pairs were analysed with Pearson correlations across all ASVs and separated by the 8 most common phyla. Lastly, we tested for differences in relative abundances of families between sample pairs, and ran paired Wilcoxon signed-rank tests on families with a mean relative abundance of >1% as relative abundances were non-normally distributed, and p-values were adjusted with Bonferroni correction for multiple testing.

Results

Preservation treatment did not bias library size

We obtained a total of 5,010,719 raw sequence reads from our 40 samples containing 1914 unique ASVs. After removing 146 contaminant ASVs from our samples, we obtained a total of 3,626,584 sequence reads and 1768 unique ASVs. The mean number of reads per sample was 90665 (range freeze-dried = 47510 to 141211, range frozen = 39323 to 183409) and treatment types did not differ significantly in number of reads per sample (p = 0.63).

Preservation treatment did not bias alpha diversity

Although there was substantial between-sample variation, the preservation treatment did not significantly bias measures of alpha diversity. For both sample treatments, we found large variation in ASV richness (freeze-died: mean = 304.55, range = 202–405; frozen: mean = 303.30, range = 202–402), Shannon index (freeze-dried: mean = 3.96, range = 3.20–4.55; frozen: mean = 3.97, range = 2.42–4.54) and Faith’s PD (freeze-dried: mean = 38.24, range = 27.41–47.63; frozen: mean = 37.13, range = 24.86–47.44) among samples. Some of that variation was explained by library size (ASV richness: R = 0.68, p = <0.001; Shannon index: R = 0.12, p = 0.48; Faith’s PD: R = 0.6, p < 0.001). However, after controlling for the effect of library size, the preservation treatment of the sample replicates did not significantly bias ASV richness, Shannon index and Faith’s PD (Fig. 1A residual ASV richness LMM (Estimate +- Std. Error): Intercept = 1.73 +- 8.09, estimate Treatment = 5.05 ± 7.542, p = 0.09; Fig. 1B Shannon index LMM: Intercept = 3.94 ± 0.10, estimate Treatment = 0.004 ± 0.07, p = 0.96; Fig. 1C residual Faith’s PD LMM: Intercept = 0.69 ± 0.98, estimate Treatment = −1.46 ± 0.87, p = 0.10), and the correlations between replicate samples were positive (Fig. 1D residual ASV richness: R = 0.64, p =0.003; Fig. 1E Shannon index R = 0.7, p < 0.001; Fig. 1F residual Faith’s PD R = 0.62, p < 0.001).

Preservation treatment has minor effect on beta diversity

Compositional differences between samples of the preservation treatments were minor. PCAs of Euclidean distances on both Hellinger and CLR transformed counts of ASVs revealed no clear clustering of sample treatment (Fig. 2). Similarly, NMDS on phylogenetic distance measures did not reveal clustering by sample treatment (Fig. 3).

Figure 1 Alpha diversity measures of freeze-dried and frozen fecal samples.

Comparison of (A) Residual ASV richness (number of ASVs), (B) Shannon index, and (C) residual Faith’s phylogenetic diversity (PD) between sample replicates from different treatments. Lines between points in A–C connect sample replicates of the same original sample identity and point colour represent the two sample preservation treatments: freeze-dried (yellow); frozen (blue-grey). Correlations (Pearson’s R) between alpha diversity measures from sample replicates of different treatments for (D) residual ASV richness, (E) Shannon index, (F) residual Faith’s PD. When excluding the outlier in (E), the results did not change qualitatively (R = 0.51, p = 0.027).

Figure 2 Clustering of freeze-dried and frozen fecal samples by Euclidean distance measures.

Principal Component Analyses (PCA) of (A) Euclidean distances of Hellinger and (B) of centered log-ratio (CLR) transformed counts. Lines between points in A and B pair replicates of the same original sample and point colour represent the two sample preservation treatments: freeze-dried (yellow); frozen (blue-grey).

Figure 3 Clustering of freeze-dried and frozen fecal samples by phylogenetic distance measures.

Non-metric multidimensional scaling (NMDS) on (A) weighted and (B) unweighted UniFrac distances. Stress weighted = 0.168, unweighted = 0.165. Outlier sample identity 4 was excluded, see Fig. S2 for NMDS on UniFrac distances including all 20 sample pairs. Lines between points in figure A and B pair replicates of the same original sample and point color represent the two sample preservation treatments: freeze-dried (yellow); frozen (blue-grey).

Consistent with the interpretation of the PCAs and the NMDS, a PERMANOVA analysis revealed that sample treatment explained only a small proportion of the variation among the samples (1.5–2.2%) whereas the sample identity was identified as the main source of variation (73.2–86.6%, Table 1). Library size was significant for Euclidean distances on CLR-transformed data and unweighted UniFrac, but only explained 1.8−3.1% (Table 1). The PERMANOVA analysis further confirmed that plate number did not significantly explain any of the variation in microbiome composition of any of the four distance measures (Table 1). Finally, we did not detect any significant differences between sample treatments with Beta dispersion tests (p = 0.08−0.94, Table 1).

Table 1 PERMANOVA results table.

Models on dissimilarity matrices of Euclidean distances (of Hellinger or centered log-ratio (CLR) transformed counts) and phylogenetic distances (Weighted and Unweighted UniFrac). Models on UniFrac distance matrices excluding outlier sample 4. p-values < 0.05 are highlighted in bold.

Permanova	Dissimilarity matrix	Factor	F	R 2	p (PERMANOVA)	p (Beta dispersion)	
1	Unweighted UniFrac	Library size	2.601	0.031	<0.001		
1	Unweighted UniFrac	Sample identity	3.437	0.732	<0.001		
1	Unweighted UniFrac	Treatment	1.549	0.018	0.03	0.937	
2	Unweighted UniFrac	Plate number	1.221	0.065	0.101	0.117	
3	Weighted UniFrac	Library size	0.813	0.004	0.536		
3	Weighted UniFrac	Sample identity	9.081	0.854	<0.001		
3	Weighted UniFrac	Treatment	4.166	0.022	<0.001	0.085	
4	Weighted UniFrac	Plate number	0.854	0.047	0.652	0.117	
5	Hellinger	Library size	1.347	0.006	0.139		
5	Hellinger	Sample identity	9.916	0.866	<0.001		
5	Hellinger	Treatment	4.737	0.022	<0.001	0.269	
6	Hellinger	Plate number	0.922	0.047	0.563	0.269	
7	CLR	Library size	2.272	0.018	0.007		
7	CLR	Sample identity	5.207	0.795	<0.001		
7	CLR	Treatment	1.926	0.015	0.021	0.563	
8	CLR	Plate number	1.128	0.057	0.235	<0.001	

Preservation treatment has a minor effect on compositional differences

Out of the 1768 ASVs, 98.5%, 91.9% and 73.7% were assigned to a phylum, family or genus, respectively. The dominating phyla were Bacteroidetes (mean relative abundance 72.9%; range 31.4–88.9%) and Firmicutes (mean relative abundance 16.4%; range 4.9–47.8%). However, within one sample, one ASV of the family Enterobacteriaceae dominated the community composition in both sample treatments with as much as 43.7% and 58.6% of the relative abundance of the freeze-dried and frozen replicates, respectively (see sample 4, Fig. S1). The ASV was not unique to the sample replicates, but prevalent in five other samples but in much lower abundance. Among the families with >1% mean relative abundance, only three families within the phylum Firmicutes, Christensenellaceae, Ruminococcaceae and Lachnospiraceae, was significantly different in relative abundances between treatments and was underrepresented within freeze-dried samples (Fig. 4, adjusted p < 0.001, Table S1).

Figure 4 Log-transformed relative abundances of families with > 1% mean relative abundance.

Amplicon Sequence Variants (ASVs) belonging to other families are combined within “family < 1% abundance”. Colour represent the two sample preservation treatments: freeze-dried (yellow); frozen (blue-grey).

About half (54%) of the ASVs were shared between the two sample treatments (Fig. S3) and these reads summed to a total of 99.5% of the reads of the full dataset. The two preservation treatment types had very similar numbers of unique ASV, and the majority of these were also unique to a single sample (Fig. S3). Overall, we found a strong correlation between relative abundances of ASVs between the two types of treatments (R = 0.87, p < 0.001). When analysing the phyla separately, the more common phyla still showed strong correlations (Fig. 5). The phyla Lentisphaerae and Spirochaetes showed weaker trends between treatment types (Fig. 5), and Lentisphaerae was the only phylum that had overrepresentation of unique ASVs in one of the sample pairs (the freeze-dried treatment, Fig. 5).

Figure 5 Correlations of relative Amplicon Sequence Variant (ASV) abundances between the two sample treatments for the eight most common phyla (freeze- dried or stored in minus 80 freezer).

The R value is the Pearson correlation across ASVs within each phylum.

Discussion

Fecal samples from wild animal populations can yield important insights into the role of the gut microbiome for the fitness, health and survival of their hosts (Suzuki, 2017). However, the transport of samples in stable cold chains from remote field collection sites to laboratories can be challenging and ways of preserving fecal samples for microbiome analyses without freezing or chemical preservatives would facilitate the development of this research field. In this study we show that frozen and freeze-dried samples do not differ in alpha diversity measures and the variation induced by the treatment amounted to less than 2.2% of the variation across the samples.

Alpha diversity measures did not differ between the two preservation methods, similar to other studies comparing alternative sample methods to deep-frozen microbiome samples (Song et al., 2016; Gavriliuc et al., 2021). Because freeze-drying did not bias any of the three alpha diversity measures we investigated, contamination or shifts in the sample community composition during the additional freeze-drying procedure are likely rare. Compared to freeze-drying, some other preservation methods that successfully eliminate the necessity of freezing sometimes come with the drawback of introducing the risk of altering sample alpha diversity through either contamination by additional bacterial exposure or change how easily different taxa are extracted. For example, Song et al. (2016) found that FTA-cards had increased Shannon diversity compared to other sample methods and fresh samples. Furthermore, the library sizes in our study did not differ systematically between the treatment types (Fig. S4) which suggests that rare ASVs are sequenced with similar probabilities (Wu et al., 2010), and both methods perform equally using the same DNA-extraction and library preparation protocol. Our data suggests that none of these possible biases were detectable in the comparison between frozen and freeze-dried samples.

For all beta diversity measures, sample preservation treatment explained a minor proportion of the variation (1.5–2.1%) within the microbiome community composition whereas sample identity explained a much larger part of the variation (73.2 –86.6%). Sample identity explained more of the variation in beta diversity calculated on Hellinger-transformed than CLR-transformed data. Hellinger transformation puts weight on abundant ASVs (Legendre & Legendre, 2012) and might hence be less sensitive to random sampling of rare ASVs which might explain the higher impact of sample identity for this transformation. Alternatively, the CLR-transformed data was better at picking up differences between pellets from the same animal as each replicate tube from an animal contained one to two pellets and replicate samples did not originate from a homogenized mix of pellets. While the variation explained by treatment was still small compared to the sample identity, it suggests that taxonomic groups are most often prevalent in both sample pairs but that some taxonomic groups vary slightly in abundance between treatments (see further discussion below). Sample library size explained a similar proportion of the variation as sample treatment, and because sample treatment did not bias library sizes, this suggest that the amount of variation caused by sample treatment was similar to random variation in library sizes between samples.

The relative abundances of ASVs were overall strongly correlated between the two treatments, suggesting that freeze-died samples reliably reflect the prevalence and abundance of bacterial populations in deep-frozen samples. Notably, one sample pair was dominated by the same single ASV in both sample treatments, showing that both treatments succeeded at capturing “extremes”. Given that this ASV had unusually high abundance within both treatments of the sample and was prevalent in other samples too, it was likely a truly dominating ASV within the gut microbiome of the individual and not a contamination. Furthermore, both treatments had similar proportions of sample-pair unique ASV abundances of the more common phyla and these were in general lower in relative abundance than ASVs shared within sample pairs, demonstrating that both treatments do equally well with sequencing less dominant ASVs. The phylum with the lowest correlation was Lentisphaerae, in which ASVs unique to one of the samples were overrepresented within freeze-dried samples (Fig. 4). Although correlations among ASVs were in general strong within the dominating phyla, we found that among families with >1% mean relative abundance, three families within the second most common phylum Firmicutes differed significantly between treatments. It is likely that differences in abundance of these taxa were responsible for the small effect of treatment on variation in in microbiome community structure. Given that this was still small compared to the effect of sample identity in all four beta diversity measures, our conclusion is that freeze-drying has only minor influences on the composition and reflect the overall diversity of the fecal microbiome well. It is however important to note that fecal microbiome samples are widely known to slightly change in composition with time and sample preservation method and the importance of not mixing different preservation methods to avoid batch effects is widely acknowledged (Song et al., 2016). Because our results suggests that there may be a minor but significant bias in the community composition of freeze dried samples of a magnitude of 1–2% percent of the variation, we suggest that researcher considering mixing both frozen and freeze dried samples should consider whether the benefits of mixing outweigh the costs of introducing this small bias.

Conclusion

Our results show that freeze-drying can be a suitable method when microbiome samples have to be exported over long-distances and stable cold chains are not available. Alpha diversity measures were not biased between frozen and freeze-dried samples and the sample preservation treatment had only a minor influence on the community composition whereas sample identity explained a large proportion of the variance. Together, our findings suggest that already collected samples stored in freezers can be thawed and freeze-dried to be transported and stored at ambient temperatures prior to 16S microbiome analyses.

Supplemental Information

Supplemental Information 1 Relative abundances of families with > 1% mean relative abundance

Sample replicates as either freeze-dried or frozen on y-axis and relative abundance on x-axis. Amplicon Sequence Variants (ASVs) belonging to other families are combined within “family < 1% abundance” and colors explained in legend.

Click here for additional data file.

Supplemental Information 2 Clustering of freeze-dried and frozen fecal samples phylogenetic distance measures including all samples

Non-metric multidimensional scaling (NMDS) on (A) weighted and (B) unweighted unifrac. Stress weighted = 7.705324e−05, unweighted = 0.1652642. Lines between points pair replicates of the same original sample and point colour represent the two sample preservation treatments: freeze-dried (yellow); frozen (blue-grey).

Click here for additional data file.

Supplemental Information 3 Number of shared and unique Amplicon Sequence Variants (ASVs) per sample treatment

Out of 1768 ASVs in our data set, 951 were shared between both sample treatments. The two treatments had a similar number of unique ASVs (417 and 400 ASVs). Grey shade of bars represents ASVs unique to a single sample (N = 735).

Click here for additional data file.

Supplemental Information 4 Rarefaction curves, each line is a sample

Sequencing depth is the number of reads within a sample and Number Amplicon Sequence Variants (ASVs) is the number of ASVs detected within the given sample. Colour of line represent the two sample preservation treatments: freeze-dried (yellow); frozen (blue-grey).

Click here for additional data file.

Supplemental Information 5 Results paired wilcoxon signed-rank tests families with mean relative abundance > 1%

p.adjust are p-values adjusted with Bonferroni adjustment for multiple testing.

Click here for additional data file.

We are grateful to Tim Clutton-Brock for his support and advice during this project. We thank Marta Manser and the Kalahari Research Trust for their tireless effort organizing the research site and we are grateful to Walter Jubber and Dave Gaynor for managing the research site. We thank Yannick Francioli, Andreas Friedrichs, Megha Rao, Johanna Sunde and Daniël Wille for help with the fieldwork. We thank the Mammals Research Institute of the University of Pretoria for their support of the research in the Kalahari.

Additional Information and Declarations

Competing Interests

Author Contributions

Animal Ethics

Data Availability

The authors declare there are no competing interests.

Hanna M. Bensch conceived and designed the experiments, performed the experiments, analyzed the data, prepared figures and/or tables, authored or reviewed drafts of the paper, and approved the final draft.

Conny Tolf performed the experiments, authored or reviewed drafts of the paper, and approved the final draft.

Jonas Waldenström conceived and designed the experiments, authored or reviewed drafts of the paper, and approved the final draft.

Daniel Lundin analyzed the data, authored or reviewed drafts of the paper, and approved the final draft.

Markus Zöttl conceived and designed the experiments, performed the experiments, authored or reviewed drafts of the paper, and approved the final draft.

The following information was supplied relating to ethical approvals (i.e., approving body and any reference numbers):

The animal ethics committee of the University of Pretoria (EC050-16).

The following information was supplied regarding data availability:

The R-scripts, asv-table and metadata are available at GitHub: https://github.com/HannaBensch/FreezeDriedVSFrozen.

The raw 16S sequences are available at NCBI short read archive (SRA): PRJNA781121.

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
