# Peer review of "Freeze-drying can replace cold-chains for transport and storage of fecal microbiome samples"

_PeerJ, doi:10.7717/peerj.13095_

## Round 0.1 · original submission · Major Revisions

The reviewers suggest some moderate changes in the manuscript, with which I am also in line. Please provide a detailed point-by-point rebuttal letter to each of the reviewers' comments, along with your revised manuscript.

Reviewer 1 ·

Basic reporting

This interesting study looks at the effect of freezedrying vs freezing on gut microbiome composition, using 16S amplicon sequencing. The authors apply 20 samples collected from Damaraland molerats, and split each sample into two treatments, generating 40 samples. The aims and language are straightforward and clear, and the overall manuscript is relatively short and to the point, which I appreciate. Since freeze-drying isn’t a standard methodology in the field, the study provides useful information on this method. Data and code are available.

My overall impression was that whilst the study is generally sound, a few more details would improve the usefulness of the paper. Given this is a PeerJ submission which focuses on analysis integrity, I also have quite a few nit-picky comments about some of the details of the analysis which should be clarified.

Conclusions/interpretations: I would suggest having more detail on how microbiome results differ between the two methods in the abstract and conclusions, rather than strongly focusing on the similarities. I think it is important to note that whilst there may be no consistent bias using one method or the other, this is not the same as no/little effect. No effect would mean no consistent bias AND that diversity metrics are perfectly correlated at the sample level. It is perfectly possible for there to be no difference between treatment groups, yet also no correlation in composition at the sample level between the two treatment groups, which would suggest that differences are random yet still potentially serious for results.

Background/Context: Whilst I think a concise introduction and discussion is generally good, I did find it rather light on details. Whilst there are only few studies looking at the effect of freezedrying on composition, there are possibly hundreds looking at how different storage methods affect measured results. I think briefly summarising what is known about this field (~one paragraph) would provide useful context. For example, it is my understanding that overall microbiome measurements seem to be rather robust to various storage methods, with things like DNA extraction kits and primer choice being more important. For someone reading this paper who is looking to decide how to store their samples, or indeed for someone who is completely new to the field, some broad context would be interesting and useful.

Different methods will always generate slightly different results, and even where effect sizes are small it is good to know exactly how they differ. Whilst there appears to be no consistent bias in a given direction by using a certain method, it is clear from the figures that at the sample level differences are generated between the two methods, but this is hardly mentioned and perhaps could give the impression that different storage methods have no effect on composition as the sample level. It seems like there are differences at the sample level, but this isn’t consistently reported for all analyses. In fact, Figure 3a shows quite large differences at the sample level, which I find rather surprising. Summarising and quantifying these differences, and whether they are due to differences in abundant/rare taxa, or perhaps specific taxonomies, would be invaluable. I have more suggestions on how to do this in the next sections.

Experimental design

Bioinformatics reporting: It would be good to see a few more details about what percentage of reads/ASVs were filtered at each major step. Particularly important is having more details on how the authors filtered contaminants and how many were filtered out, which currently is very vague. It is possible to have reverse contamination, where ‘real’ ASVs get into the negative controls. R packages like ‘decontam’ are made make this process a bit more reproduceable, but do depend on having multiple negative controls. Filtering out contaminants can be tricky and in my experience lab contaminants shouldn’t really account for more that ~0.1% of reads. Either way, these steps and how many ASVs were filtered out should be reported. In addition, I think how many ASVs are filtered at rarefying should also be reported.

Alpha diversity analysis: The counts were rarefied, which I don’t have a problem with, but it is more than possible that even after rarefaction there is a signal of sequencing depth, since rarefying is sometimes not enough to completely control for this bias. This should be reported, and if the effect is still there, controlled for.

Alpha diversity analysis: The study reports the effect of treatment on alpha diversity, but not the effect of sample ID, which would also be useful to know (this should effectively represent the correlation between the two treatments at the sample level, and is represented by the model ICC). This can be easily extracted from the GLMM using performance::icc(model, by_group = TRUE). Personally I would prefer Fig. 1 to show this correlation (opposed to a boxplot), with linear trends for the two treatment groups. I think this would be a more effective way of showing the effects at both the treatment level and the sample level together. If there is a low correlation, this would be a bit worrying, but it’s hard to tell from the data presented. Depending on this correlation, it may be required to rethink the subtitle “Preservation treatment did not affect alpha diversity”. As mentioned, there may be no consistent bias, but that does not mean there is no effect.

Beta diversity analyses: The normalization protocol for the beta diversity analyses is not completely clear, and there is perhaps some confusion regarding differences in transformations, dissimilarity indices, ordinations, and permanovas, which is quite common as it can be quite confusing. Just to be clear, the first two beta diversity analyses perform count transformations, but it is not clear which dissimilarity index is used to generate the distance matrix. In vegan/phyloseq, I think Bray-Curtis is the default index if not specified. Hellinger can be a transformation (sqrt of relative abundance) or a dissimilarity index, confusingly, but I dont think it is offered by vegan. I assume in this case it is a transformation. In the second two beta diversity analyses, the two dissimilarity indices are reported (Weighted and Unweighted Unifrac), but it is not clear how the data were normalised/transformed (rarefied? CLR? Hellinger?). On top of this, for visualising beta diversity, the first two plots use PCA (data reduction into n-1 axes), whilst the first two plots use NMDS (which presumably generates 2 axes, although it is not explicitly reported). As such, beta analyses 1 and 2 differ in so many ways from analyses 3 and 4 that it is difficult to gauge where any differences stem from. Perhaps some careful consideration is needed regarding what exactly the authors want to test, then change one thing (either transformation, dissimilarity index, type of ordination) and compare. The type of ordination makes no difference to the PERMANOVA, since this is of course based on the dissimilarity matrix and not the ordination, but it will make a difference to the plots. Overall, each analyses needs a clear description of 1) the data transformation, and 2) the dissimilarity index used to generate the distance matrix.

Adonis: The authors do not explicitly state how they set up the PERMANOVA, which is sensitive to variable order unless the ‘by’ argument is stated as “marginal”. If the authors are measuring marginal effects, this should be reported. If they are not, then the order of the terms can be important, and worth checking whether entering the terms in a different order affects anything. In addition, please include the effect size (F) in Table 1. The R2 and F represent different things, and R2 isn’t exactly the effect size but rather explanatory power (effect size in this case = how far centroids are from one another).

Permanova results: That sample ID only explains ~60% of variation in one analysis is surprising. This means there is nearly 35-40% unexplained variation between the same sample when using the two different methods. For me this is quite a lot. A key question for me is whether this is variation in rare taxa or abundant taxa? Can it be explained by differences in sequencing depth? This is worth adding to the model too to see if it can mop up any of that unexplained variation. Might be worth also testing out different normalisation methods for this analysis, since one ideally wants a normalisation method that maximises the effect of sample identity.

Differential abundance analysis: how were the data normalised for this analysis?

Overall the experimental design can answer the main questions. My one query was why the length of time a sample had been freeze-dried or frozen for was not incorporated into analyses, at least as a check. I wonder whether samples that had been freezedried for longer tended to be more different from their frozen counterparts than samples that were more recently freezedried, given these may be expected to be more degraded. Is there any evidence for this?

Validity of the findings

Please see comments above about summarising the sample-level differences rather than just the treatment-level differences.

Reviewer 2 ·

Basic reporting

The submitted manuscript is professionally written, with a very clear and understandable message. It was a pleasure reviewing such a well-structured and nicely flowing paper. The authors have done a fantastic job highlighting the research gap and putting their work into context of the existing literature. The presented work is an important step towards enabling future microbiome studies in remote regions and has the potential to be expand to other animals as well as humans. In particular, the authors have performed a very thorough data and statistical analysis. The only technical limitation is the sample storage in a super freezer prior to freeze drying, which is exactly the requirement the authors attempted to eliminate in this study. This is likely due to the remote study location and hopefully can be overcome in future studies.

As required by the journal (“Uncommon abbreviations should be spelled out at first use”), please spell out ASV in the abstract. Although this is becoming a more commonly used abbreviation, it is not yet universally used outside of the field.

All raw data files are depicted appropriately and are accessible within the manuscript
submission files or the respective data repositories. However, the deposition information is not noted in the manuscript. Please rectify this by inserting a data deposition and availability statement within the Methods section to comply with the requirements of the journal. When doing so please include the specific sample accession numbers for ease of the reader. Otherwise, it appears difficult to access the described samples without having to search through hundreds of samples in the BioProject PRJNA781121.

Line 64-67: “Although freeze-drying has rarely been used in microbiome studies (Gavriliuc et al., 2021), it offers considerable advantages; no chemicals are added and the samples can be stored and transported at ambient temperature, eliminating the need for cold chain logistics and hazardous material handling.” Instead of such a complex, long sentence, I suggest removing the semicolon and simply splitting this statement into two sentences.

Typically, abbreviations as part of figure headers and legends are spelled out again at first mentioned in order for the reader to understand the figure as a stand-alone without the main manuscript text.

As required by the journal (“When citing use the abbreviation 'Fig.'. When starting a sentence with a citation, use 'Figure 1'.”) please adjust your figure reference to adhere to the guidelines. Furthermore, the figures are not sorted according to their numbering in the text. Your first figure reference is Figure 3 (Line 163 in text) prior to Figure 1 (Line 197). Please either relabel the figures accordingly or switch the figures. Please also adapt all figure titles and legends to adhere to the journal’s instructions, which are for example for Figure 1 as follows: Figure 1: Alpha diversity measures of freeze-dried and frozen fecal samples. A) ASV richness (number of ASVs), B) Shannon index, C) Faith’s PD. Lines between points connect sample replicates of the same original sample identity and point color represent the two sample preservation treatments: yellow freeze-dried; blue-grey frozen.

Experimental design

An appropriate ethical approval statement is provided by the authors as part of their Methods section.

Line 40: “Super freezers require a reliable source of electricity and are expensive to purchase and maintain.” Although this statement is factual correct, freeze driers advocated as the superior piece of equipment, also require a reliable source of electricity. How feasible is it to integrate this requirement considering the remote location of the field site? The manuscript would benefit from including a section on the feasibility of this requirement.

Lines 47-50: "Transportation of frozen samples hence introduces the risk of thawing and incurs additional costs of cold chain logistics. This is especially problematic when the sampling location is in a remote area or in another country than the laboratory facility and commercial couriers are needed.” This section appears a little repetitive and could likely be reduced to one sentence with a more concise message.

Line 79: In the methods section the field work time frame is stated from May to October 2019, however the online metadata sheet states April as a collection date. Please explain this discrepancy and adjust accordingly.

Line 79: Seeing that the samples were collected over a time frame from April/May to October 2019 what impact do seasonal food sources have on the gut microbiome? Has this factor been investigated previously or could you comment on food availability in this region? Further, since animals from the same families have been used for this study, have comparisons between and within families been performed? This factor can have implications on the microbiome analysis as the vaginal microbiota impacts overall animal health from birth.

Lines 85-87: “Freeze-dried samples were then stored and transported at room temperature while frozen replicates were transported on dry ice with a commercial courier service to Sweden.” It is stated that freeze dried samples have been stored at room temperature which is likely incorrect due to temperature variations between the African climate, the cold air during shipping, and storage at final destination. Ambient temperature would likely be more appropriate to use in this instance. Could you highlight how changes in temperature can impact sample composition?

Lines 82-83: “Each sample was split into two replicate tubes with 1-3 fecal pellets per tube and stored in a -80°C freezer at the field site”. Please comment on the fact that even though this study is focused on replacing the requirement for super freezers the samples were stored in a super freezer prior to freeze drying. This statement implies that there is still a need for super freezer immediately upon sample collection and also a potential impact of freezing with subsequent freeze drying on the fecal microbiome composition. This study limitation should be highlighted more clearly in the discussion section with suggestion on how to overcome it where appropriate.

Line 83: “After all samples had been collected...” Could you please elaborate on the study timepoints and include approximate timeframes in your manuscript (e.g. a flowchart)? If some samples were collected on the 1st of May, were they stored in a super freezer until October (last collection day) before being freeze dried? And how would this impact the microbiome composition? Has a comparison been made between samples stored for months prior to freeze drying and those freeze dried immediately?

Lines 140-141: “We combined reads of samples on plates that had been sequenced twice (plate 2 and 3).” Could you please clarify and add if all samples were sequenced in duplicate or only a fraction and if so, why were not all sequenced twice?

Figure 2: Can you comment on the clustering effect seen in Figure 2? Are these animals from the same region or family for example?

Metadata: For ease of the reader and to support open access practices such as replication, it would be beneficial to include the metadata file for all animals in this study as part of the supplementary files. Furthermore, the metadata should be described in more detail to enable interpretation and replication of the results e.g. (currently missing) collection date, transport and storage timeframe, DNA extraction date, family number, longitude, latitude etc.).

Validity of the findings

Line 279: Please elaborate on the cause of significant differences between plates. Typically, technical batch effects are seen across sequencing runs, personnel, or kits (Wang & LêCao 2019, https://doi.org/10.1093/bib/bbz105). Assuming the same protocol, reagents and personnel were used for sample preparation, differences between 96-well plates are not expected and might warrant discussion in the discussion section of the manuscript.

Additional comments

Lines 161-162: “figure S1… Figure S2” Please be consistent with wording and referencing of figures throughout the manuscript according to the journal guidelines.

Raw data files: From the raw data files, it appears that the samples used are part of a larger study. The sample numbers indicate a spread over hundreds of samples. Could you please comment on and include the selection criteria for the here used samples within the manuscript as this might cause potential sample bias? Were these selected randomly from the overarching study, according to a different criterion e.g., first 20 animals caught, age, family etc. or in retrospect after sequencing?

Table 1: “The stars indicate level of statistical significance.” Please include the significance levels in the table legend seeing that p-value levels can differ between publications and to aid the reader’s understanding.

Table S1: The supplementary table is missing a table header as well as appropriate description. Please include these.

·

Basic reporting

· L65: Though it was noted that freeze-drying has not been widely used, there have been a handful of studies that appear to have used this approach (at least within the past year, for example https://doi.org/10.1016/j.medmic.2021.100044 or10.1080/19490976.2020.1759489 ). To be comprehensive, I think it would be helpful to briefly mention these in the introduction or whether they support/contradict findings in the discussion.
· L95: I accessed the SRA data from the larger microbiome study, but I am not sure which samples were used for the present study. Can a list be provided either in-text or as part of some supplementary material?
· L96: Can the authors provide some information on the storage length for each sample? Are storage times among these samples evenly represented between 4 - 10 months or were most samples stored closer to 4/10 months, either in the text or in supplementary material? Also, can or have the authors tested for the proportion of variance explained by storage duration? I am wondering if samples stored for say the maximum 10 months show differences from those stored at 4 etc (I do not think this needs to be changed in the manuscript, just a confirmation in the author response).
· L124 - 131: It was said that the Ampliseq workflow was followed. Were there any other steps between running cutadapt and the denoising step? And were all steps/commands run with the same parameters as the workflow/all default parameters? Please clarify.
· L160 - 163: Please report the stress value of the NMDS before and after removing the outlier samples as well.
· L183: How was this comparison done? I am assuming with a t-test? If not described in the methods please indicate.

Experimental design

· L83: I am not familiar with fecal samples from mole rats. Could the sample have been split into multiple aliquots for each treatment to test for variation within a sample and treatment? It is too late to change this now, but my thinking is that some of the variation we could be seeing between treatments could be from variation within a treatment.

Validity of the findings

· L83 - 87: My main concern is that, similar to other studies that have compared storage methods, there was no comparison of the stored samples versus samples that were immediately extracted. As such, there is no way to confirm that the different treatments reflect the microbial community from when the samples were obtained, and therefore the only conclusions that can be drawn are that the community structure did not differ between treatments. While this may have been the intended goal of the present study, I think it is important to briefly address this limitation somewhere in the text.
· L195 - L196: The authors mention “preservation treatment of sample replicates did not significantly affect ASV richness”. Though this is supported by the LMM, and I agree with these conclusions, one sample in Figure 1A has approximately 100 ASVs when freeze-dried and approximately double that when frozen. Can the authors provide a possible explanation?

Additional comments

The following are minor comments pertaining to grammar and punctuation that the authors may consider for the improvement of their manuscript.
· L56 - L57: Consider changing “and has been suggested to preserve the microbial community the best” to “and has been shown to be the most effective for preserving microbiome community structure”
· L144 - 145: Change “were calculated on subsampled and rarefied ASV counts to the minimum library size” to “were calculated on subsampled ASV counts rarefied to the minimum library size”
· L147: Change “analyzed with Linear mixed models” to “analyzed using linear mixed models”
· L154 - 155: Change “Hellinger transformation” to “The Hellinger transformation” and change “while CLR” to “while the CLR transformation”
· L157 - L158: Add the word distances to “we calculated weight and unweighted UniFrac distances”
· L161 (and throughout): italicize taxonomic names like Proteobacteria
· L167 - 168: “as strata argument” to “as the strata argument”
· L211 (and throughout): I believe “table 1” should be capitalized to “Table 1”?
· L213 - 214: change “did not significantly explain any of the variation” to something along the lines of “the proportion of variation explained by plate number was low/negligible and was not significant”
· L271: Change to “explained more of the variation in Hellinger-transformed data than in CLR-transformed data”.
· L274 - 275: Change to “were the only distance measures with a significant effect by treatment”.
· L278 - L279: Comma after “Compared to the Euclidean distance measures”
· L294: Remove comma after “likely”
· L307: change to “be exported over long distances”
· L311: change to “stored at ambient temperatures”
· Figure 2: Consider putting the color of the treatment in parentheses at the end of the caption. “freeze-dried (yellow); frozen (blue-grey)”.

---

## Round 0.2 · Minor Revisions

The paper requires a few minor amendments before finalizing its acceptance, which I hope you will be able to address fairly soon.

Reviewer 1 ·

Basic reporting

The authors have addressed all my previous comments and I am happy for the paper to be published. Please note that Fig1 d-f do not appear to be referenced in the main text.

Experimental design

No comment.

Validity of the findings

No comment.

Reviewer 2 ·

Basic reporting

no comment

Experimental design

no comment

Validity of the findings

no comment

Additional comments

The authors have addressed all previous comments thoroughly and extensively in their rebuttal, which has increased the manuscript's quality to meet all journal criteria. Therefore, I recommend this article to be accepted for publication.

·

Basic reporting

· I commend the authors for their rapid turnaround and ability to address all reviewer comments.
· I also commend the authors for their analysis and documentation, which was reproducible. Using information from the methods section, I was able to run the AmpliSeq workflow and produce an ASV table that was nearly identical to that posted on Github. Only minor differences in ASV abundances were noted. Could this be due to the use of a seed number used to generate the ASV table? If so, could this number be included somewhere in the manuscript?
· I was also able to re-create almost every figure. The only figure I could not re-create required the use of the “pseq.rds” object, which was not available on the Github repository, making the “FDvsfrozen_permanovatable.R” script unable to run. Can this object or the code used to generate this object be included somewhere in the Github repo?
· Some more minor comments for making the analysis more reproducible: I often had to change the directory for data files from “../data/” to “datafiles/” or “../datafiles/” depending on which script I was using, and the .tree file was in the datafiles folder instead of the data/seff/ folders as was in the scripts. I leave this up to the authors whether they would like to change this or not as it was not a major concern or problem.
· There are two scripts titled “Freezedried_vs_min80_relab.Rmd” in the repository, one contained in the main folder and one contained in the scripts folder. Is there a difference between these two scripts?

Experimental design

No comment

Validity of the findings

· L282 - 284: Though sample preservation treatment only explained 1.5 - 2.1% of variation, this can be a substantial amount depending on the aims of the study. I would suggest changing the wording to reflect that the proportion of variation explained by preservation treatment was minor relative to other factors like sample identity.
· L320 - 321 and 329 - 331: I am a little confused by the wording here. First the authors state that preservation methods should not be mixed, and then it is said that samples that are currently frozen can be thawed? This could possibly be interpreted as ‘mixing preservation methods’. My understanding was that, from this study, we can conclude that any new samples can be freeze-dried and stored at ambient temperatures, but the authors did not test whether freezing, then thawing + freeze-drying yields similar community composition to samples that are just frozen or just freeze-dried. I think it would be helpful to clarify the final sentence in the conclusion prior to publication.

Additional comments

· L291: “origin” should be “originate”
· L311: please add a comma after “1% mean relative abundance” to improve readability

---

## Round 0.3 · accepted · Accept

Thank you for your fast and thorough revision. Looking forward to your future submissions in PeerJ!